META-RESEARCH

# Dataset decay and the problem of sequential analyses on open datasets

**Abstract** Open data allows researchers to explore pre-existing datasets in new ways. However, if many researchers reuse the same dataset, multiple statistical testing may increase false positives. Here we demonstrate that sequential hypothesis testing on the same dataset by multiple researchers can inflate error rates. We go on to discuss a number of correction procedures that can reduce the number of false positives, and the challenges associated with these correction procedures.

**WILLIAM HEDLEY THOMPSON\*, JESSEY WRIGHT, PATRICK G BISSETT AND RUSSELL A POLDRACK**

**\*For correspondence:** william. thompson@stanford.edu

**Competing interests:** The authors declare that no competing interests exist.

## Introduction

In recent years, there has been a push to make the scientific datasets associated with published papers openly available to other researchers (*Nosek et al., 2015*). Making data open will allow other researchers to both reproduce published analyses and ask new questions of existing datasets (*Molloy, 2011*; *Pisani et al., 2016*). The ability to explore pre-existing datasets in new ways should make research more efficient and has the potential to yield new discoveries (*Weston et al., 2019*).

The availability of open datasets will increase over time as funders mandate and reward data sharing and other open research practices (*McKiernan et al., 2016*). However, researchers re-analyzing these datasets will need to exercise caution if they intend to perform hypothesis testing. At present, researchers reusing datasets tend to correct for the number of statistical tests that they perform on the datasets. However, as we discuss in this article, when performing hypothesis testing it is important to take into account all of the statistical tests that have been performed on the datasets.

A distinction can be made between *simultaneous* and *sequential* correction procedures when correcting for multiple tests. Simultaneous procedures correct for all tests at once, while

sequential procedures correct for the latest in a non-simultaneous series of tests. There are several proposed solutions to address multiple sequential analyses, namely $\alpha$-*spending* and $\alpha$-*investing* procedures (*Aharoni and Rosset, 2014*; *Foster and Stine, 2008*), which strictly control false positive rate. Here we will also propose a third, $\alpha$-*debt*, which does not maintain a constant false positive rate but allows it to grow controllably.

Sequential correction procedures are harder to implement than simultaneous procedures as they require keeping track of the total number of tests that have been performed by others. Further, in order to ensure data are still shared, the sequential correction procedures should not be antagonistic with current data-sharing incentives and infrastructure. Thus, we have identified three desiderata regarding open data and multiple hypothesis testing:

### Sharing incentive
Data producers should be able to share their data without negatively impacting their initial statistical tests. Otherwise, this reduces the incentive to share data.

Minimal to no restrictions should be placed on accessing open data, other than those necessary to protect the confidentiality of human subjects. Otherwise, the data are no longer open.

*Stable false positive rate*

The false positive rate (i.e., type I error) should not increase due to reusing open data. Otherwise, scientific results become less reliable with each reuse.

We will show that obtaining all three of these desiderata is not possible. We will demonstrate below that the current practice of ignoring sequential tests leads to an increased false positive rate in the scientific literature. Further, we show that sequentially correcting for data reuse can reduce the number of false positives compared to current practice. However, all the proposals considered here must still compromise (to some degree) on one of the above desiderata.

## An intuitive example of the problem

Before proceeding with technical details of the problem, we outline an intuitive problem regarding sequential statistical testing and open data. Imagine there is a dataset which contains the variables ($v_1$, $v_2$, $v_3$). Let us now imagine that one researcher performs the statistical tests to analyze the relationship between $v_1 \sim v_2$ and $v_1 \sim v_3$ and decides that a $p < 0.05$ is treated as a positive finding (i.e. null hypothesis rejected). The analysis yields p-values of $p = 0.001$ and $p = 0.04$ respectively. In many cases, we expect the researcher to correct for the fact that two statistical tests are being performed. Thus, the researcher chooses to apply a Bonferroni correction such that p < 0.025 is the adjusted threshold for statistical significance. In this case, both tests are published, but only one of the findings is treated as a positive finding.

Alternatively, let us consider a different scenario with sequential analyses and open data. Instead, the researcher only performs one statistical test ($v_1 \sim v_2$, p = 0.001). No correction is performed, and it is considered a positive finding (i.e. null hypothesis rejected). The dataset is then published online. A second researcher now performs the second test ($v_1 \sim v_3$, p = 0.04) and deems this a positive finding too because it is under a $p < 0.05$ threshold and they have only performed one statistical test. In this scenario, with the same data, we have two published positive findings compared to the single positive finding in the previous scenario. Unless a reasonable justification exists for this difference between the two scenarios, this is troubling.

What are the consequences of these two different scenarios? A famous example of the consequences of uncorrected multiple simultaneous statistical tests is the finding of fMRI BOLD activation in a dead salmon when appropriate corrections for multiple tests were not performed (*Bennett et al., 2010*; *Bennett et al., 2009*). Now let us imagine this dead salmon dataset is published online but, in the original analysis, only one part of the salmon was analyzed, and no evidence was found supporting the hypothesis of neural activity in a dead salmon. Subsequent researchers could access this dataset, test different regions of the salmon and report their uncorrected findings. Eventually, we would see reports of dead salmon activations if no sequential correction strategy is applied, but each of these individual findings would appear completely legitimate by current correction standards.

We will now explore the idea of sequential tests in more detail, but this example highlights some crucial arguments that need to be discussed. Can we justify the sequential analysis without correcting for sequential tests? If not, what methods could sequentially correct for the multiple statistical tests? In order to fully grapple with these questions, we first need to discuss the notion of a *statistical family* and whether sequential analyses create new families.

## Statistical families

A family is a set of tests which we relate the same error rate to (familywise error). What constitutes a family has been challenging to precisely define, and the existing guidelines often contain additional imprecise terminology (e.g. *Cox, 1965*; *Games, 1971*; *Hancock and Klockars, 1996*; *Hochberg and Tamhane, 1987*; *Miller, 1981*). Generally, tests are considered part of a family when: (i) multiple variables are being tested with no predefined hypothesis (i.e. exploration or data-dredging), or (ii) multiple pre-specified tests together help support the same or associated research questions (*Hancock and Klockars, 1996*; *Hochberg and Tamhane, 1987*). Even if following these

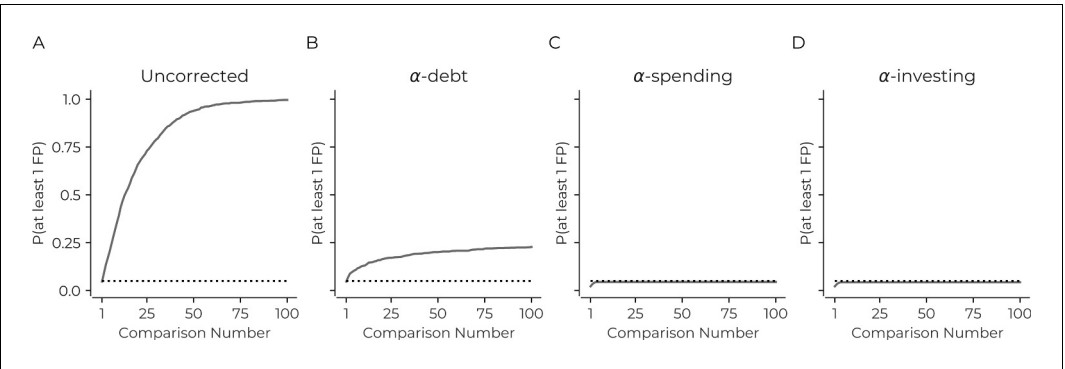

**Figure 1.** Correction procedures can reduce the probability of false positives. (**A**) The probability of there being at least one false positive (y-axis) increases as the number of statistical tests increases (x-axis). The use of a correction procedure reduces the probability of there being at least one false positive (B: α-debt; C: α-spending; D: α-investing). Plots are based on simulations: see main text for details. Dotted line in each panel indicates a probability of 0.05.

guidelines, there can still be considerable disagreements about what constituents a statistical family, which can include both very liberal and very conservative inclusion criteria. An example of this discrepancy is seen in using a factorial ANOVA. Some have argued that the main effect and interaction are separate families as they answer 'conceptually distinct questions' (e.g. page 291 of *Maxwell and Delaney, 2004*), while others would argue the opposite and state they are the same family (e.g. *Cramer et al., 2016*; *Hancock and Klockars, 1996*). Given the substantial leeway regarding the definition of family, recommendations have directed researchers to define and justify their family of tests a priori (*Hancock and Klockars, 1996*; *Miller, 1981*).

A crucial distinction in the definition of a family is whether the analysis is confirmatory (i.e. hypothesis-driven) or exploratory. Given issues regarding replication in recent years (*Open Science Collaboration, 2015*), there has been considerable effort placed into clearly demarcating what is exploratory and what is confirmatory. One prominent definition is that confirmatory research requires preregistration before seeing the data (*Wagenmakers et al., 2012*). However, current practice often involves releasing open data with the original research article. Thus, all data reuse may be guided by the original or subsequent analyses (a HARKing-like problem where methods are formulated after some results are known [*Button, 2019*]). Therefore, if adopting this prominent definition of confirmatory research (*Wagenmakers et al., 2012*), it follows that any reuse of open data after publication must be exploratory unless the analysis is preregistered before the data release.

Some may find *Wagenmakers et al., 2012* definition to be too stringent and instead would rather allow that confirmatory hypotheses can be stated at later dates despite the researchers having some information about the data from previous use. Others have said confirmatory analyses may not require preregistrations (*Jebb et al., 2017*) and have argued that confirmatory analyses on open data are possible (*Weston et al., 2019*). If analyses on open data can be considered confirmatory, then we need to consider the second guideline about whether statistical tests are answering similar or the same research questions. The answer to this question is not always obvious, as was highlighted above regarding factorial ANOVA. However, if a study reusing data can justify itself as confirmatory, then it must also justify that it is asking a 'conceptually distinct question' from previous instances that used the data. We are not claiming that this is not possible to justify, but the justification ought to be done if no sequential correction is applied as new families are not created just because the data is being reused (see next section).

We stress that our intention here is not to establish the absolute definition of the term *family*; it has been an ongoing debate for decades, which we do not intend to solve. We believe our upcoming argument holds regardless of the definition. This section aimed to provide a working definition of family that allows for both small and large families to be justified. In the next section, we argue that regardless of the specific definition of family, sequential testing by itself does not create a new family by virtue of it being a sequential test.

## Families of tests through time

The crucial question for the present purpose is whether the reuse of data constitutes a new family of tests. If data reuse creates a new family of tests, then there is no need to perform a sequential correction procedure in order to maintain control over familywise error. Alternatively, if a new family has not been created simply by reusing data, then we need to consider sequential correction procedures.

There are two ways in which sequential tests with open data can differ from simultaneous tests (where correction is needed): a time lag between tests and/or different individuals performing the tests. Neither of these two properties is sufficient to justify the emergence of a new family of tests. First, the temporal displacement of statistical tests can not be considered sufficient reason for creating a new family of statistical tests, as the speed with which a researcher analyzes a dataset is not relevant to the need to control for multiple statistical tests. If it were, then a simple correction procedure would be to wait a specified length of time before performing the next statistical test. Second, it should not matter who performs the tests; otherwise, one could correct for multiple tests by crowd-sourcing the analysis. Thus if we were to decide that either of the two differentiating properties of sequential tests on open data creates a new family, undesirable correction procedures would be allowable. To prevent this, statistical tests on open data, which can be run by different people, and at different times, can be part of the same family of tests. Since they can be in the same family, sequential tests on open data need to consider correction procedures to control the rate of false positives across the family.

We have demonstrated the possibility that families of tests can belong to sequential analyses. However, in practice, when does this occur? The scale of the problem rests partly in what is classed as an exploratory analysis or not. If all data reuse is considered part of the same family due to it being exploratory, this creates a large family. If however, this definition is rejected, then it depends on the research question. Due to the fuzzy nature of 'family', and the argument above showing that data reuse does not create new families automatically, we propose a simple rule-of-thumb: if the sequential tests would be considered within the same family if performed simultaneously, then they are part of the same family in sequential tests. The definition of exploratory analyses and this rule-of-thumb indicate that many sequential tests should be considered part of the same family when reusing open data. We, therefore, suggest that researchers should apply corrections for multiple tests when reusing data or provide a justification for the lack of such corrections (as they would need to in the case of simultaneous tests belonging to different families).

## The consequence of not taking multiple sequential testing seriously

In this section, we consider the consequences of uncorrected sequential testing and several procedures to correct for them. We start with a simulation to test the false positive rate of the different sequential correction procedures by performing 100 sequential statistical tests (Pearson correlations) where the simulated covariance between all variables was 0 (see Methods for additional details). The simulations ran for 1000 iterations, and the familywise error was calculated using a two-tailed statistical significance threshold of $p < 0.05$.

We first consider what happens when the sequential tests are uncorrected. Unsurprisingly, the results are identical to not correcting for simultaneous tests (*Figure 1A*). There will almost always be at least one false positive any time one performs 100 sequential analyses with this simulation. This rate of false positives is dramatically above the desired familywise error rate of at least one false positive in 5% of the simulation's iterations: uncorrected sequential tests necessarily lead to more false positives.

To correct for this false positive increase, we consider several correction procedures. The first sequential procedure we consider is *α-debt*. For the ith sequential test, this procedure considers there to be *i* tests that should be corrected. This procedure effectively performs a Bonferroni correction – i.e. the threshold of statistical significance becomes $\frac{\alpha_1}{i}$ where $\alpha_1$ is the first statistical threshold (here 0.05). Thus, on the first test $\alpha_1 = 0.05$, then on the second sequential test $\alpha_2 = 0.025$, $\alpha_3 = 0.0167$, and so on. While each sequential test is effectively a Bonferroni correction considering all previous tests, this does not retroactively change the inference of any previous statistical tests. When a new test is performed, the previous test's $\alpha$ is now too lenient considering all the tests that have been performed. Thus, when considering all tests together, the false positive rate will increase,

accumulating a false positive 'debt'. This debt entails that the method does not ensure the type I error rate remains under a specific value, instead allows it to controllably increase under a 'debt ceiling' with each sequential test (the debt ceiling is the sum of all $\alpha_1$ to $\alpha_t$ at $t$). The debt ceiling will always increase, but the rate of increase in debt slows down. These phenomena were confirmed in the simulations (*Figure 1B*). Finally, the method can mathematically ensure that the false negative rate (i.e., type II error) is equal to or better than simultaneous correction with Bonferroni (See Methods).

The next two procedures we consider have previously been suggested in the literature $\alpha$-spending and $\alpha$-investing (*Aharoni and Rosset, 2014*; *Foster and Stine, 2008*). The first has a total amount of '$\alpha$ wealth', and the sum of all the statistical thresholds for all sequential tests can never exceed this amount (i.e., if the $\alpha$ wealth is 0.05 then the sum of all thresholds on sequential tests must be less than 0.05). Here, for each sequential test, we spend half the remaining wealth (i.e., $\alpha_1$ is 0.025, $\alpha_2$ is 0.0125, and so on). In the simulations, the sequential tests limit the probability of there being at least one false positive to less than 0.05 (*Figure 1C*). Finally, $\alpha$-investing allows for the significance threshold to increase or decrease as researchers perform additional tests. Again there is a concept of $\alpha$-wealth. If a test rejects the null hypothesis, there is an increase in the remaining $\alpha$-wealth that future tests can use and, if the reverse occurs, the remaining $\alpha$-wealth decreases (see methods). $\alpha$-investing ensures control of the false discovery rate at an assigned level. Here we invest 50% of the remaining wealth for each statistical test. In the simulations, this method also remains under 0.05 familywise error rate as the sequential tests increase (*Figure 1D*).

The main conclusion from this set of simulations is that the current practice of not correcting for open data reuse results in a substantial increase in the number of false positives presented in the literature.

## Sensitivity to the order of sequential tests

The previous simulation did not consider any true positives in the data (i.e. cases where we should reject the null hypothesis). Since the statistical threshold for significance changes as the number of sequential tests increases, it becomes crucial to evaluate the sensitivity of each method

to both type I and type II errors in regards to the order of sequential tests. Thus, we simulated true positives (between 1-10) where the covariance of these variables and the dependent variable were set to $p$ ($p$ ranged between 0 and 1). Further, $\lambda$ controlled the sequential test order determining the probability that a test was a true positive. When $\lambda$ is positive, it entails a higher likelihood that earlier tests will be one of the true positives (and vice versa when $\lambda$ was negative; see methods). All other parameters are the same as the previous simulation. Simultaneous correction procedures (Bonferroni and FDR) of all 100 tests were also included to contrast the different sequential procedures to these methods.

The results reveal that the order of the tests is pivotal for sequential correction procedures. Unsurprisingly, the uncorrected and simultaneous correction procedures do not depend on the sequential order of tests (*Figure 2ABC*). The sequential correction procedures all increased their true positive rate (i.e., fewer type II errors) when the true positives were earlier in the analysis order (*Figure 2A*). We also observe that $\alpha$-debt had the highest true positive rate of the sequential procedures and, when the true positives were later in the test sequence, performed on par with Bonferroni. Further, when the true positives were earlier, $\alpha$-debt outperformed Bonferroni at identifying them. $\alpha$-investing and $\alpha$-spending cannot give such assurances when the true positives are later in the analysis sequence (i.e. $\lambda$ is negative) there is less sensitivity to true positives (i.e. type II errors). $\alpha$-debt is more sensitive to true positives compared to $\alpha$-spending because the threshold for the mth sequential test decreases linearly in $\alpha$-debt and exponentially in $\alpha$-spending. This fact results in a more lenient statistical threshold for $\alpha$-debt in later sequential tests.

The false positive rate and false discovery rate are both very high for the uncorrected procedure (*Figure 2BC*). $\alpha$-debt and $\alpha$-spending both have a decrease in false positives and false discovery rate when $\lambda$ is positive (*Figure 2BC*). The false discovery rate for $\alpha$-debt generally lies between the spending (smallest) and investing procedures (largest and one that aims to be below 0.05). Also, for all methods, the true positive rate breaks down as expected when the covariance between variables approaches the noise level. Thus we split the false discovery rate along four quadrants based on $\lambda$ and the noise floor (*Figure 2D*). The quadrants where true positive covariance is above the noise floor (Q1

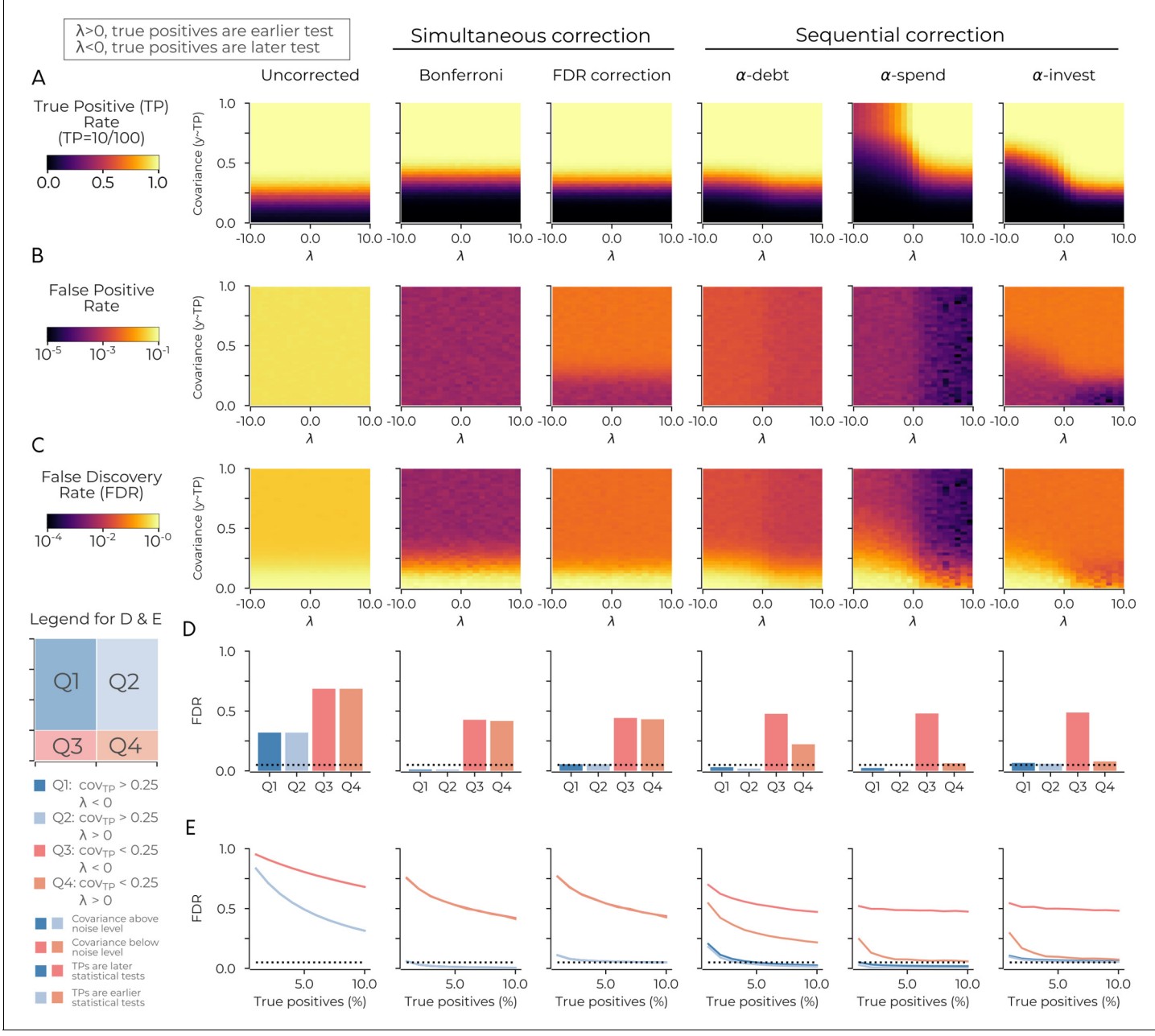

**Figure 2.** The order of sequential tests can impact true positive sensitivity. (**A**) The true positive rate in the uncorrected case (left-most panel), in two cases of simultaneous correction (second and third panels), and in three cases of sequential correction (fourth, fifth and sixth panels). In each panel the true positive rate after 100 tests is plotted as a function of two simulation parameters: $\lambda$ (x-axis) and the simulated covariance of the true positives (y-axis). When $\lambda$ is positive (negative), it increases the probability of the true positives being an earlier (later) test. Plots are based on simulations in which there are ten true positives in the data: see main text for details. (**B**) Same as A for the false positive rate. (**C**) Same as A for the false discovery rate. (**D**) Same as C for the average false discovery rate in four quadrants. Q1 has $\lambda < 0$; covariance $> 0.25$. Q2 has $\lambda > 0$; covariance $> 0.25$. Q3 has $\lambda < 0$; covariance $< 0.25$. Q4 has $\lambda > 0$; covariance $< 0.25$. The probability of true positives being an earlier test is highest in Q2 and Q4 as $\lambda > 0$ in these quadrants. (**E**) Same as D with the false discovery rate (y-axis) plotted against the percentage of true positives (x-axis) for the four quadrants. The dotted lines in D and E indicate a false discovery rate of 0.05. Code is available at https://github.com/wiheto/datasetdecay (**Thompson, 2020**; copy archived at https://github.com/elifesciences-publications/datasetdecay).

and Q2) has a false discovery rate of less than 0.05 for all procedures except uncorrected (**Figure 2D**). Finally, when varying the number of true positives in the dataset, we found that Q1 and Q2 generally decrease as the number of true positives grows for $\alpha$-spending and $\alpha$-debt,

whereas $\alpha$-investing remains the 0.05 mark regardless of the number of true positives (*Figure 2E*).

All three sequential correction procedures performed well at identifying true positives when these tests were made early on in the analysis sequence. When the true positive tests are performed later, $\alpha$-debt has the most sensitivity for true positives and $\alpha$-investing is the only procedure that has a stable false discovery rate regardless of the number of true positives (the other two methods appear to be more conservative). The true positive sensitivity and false discovery rate of each of the three sequential correction methods considered depend on the order of statistical tests and how many true positives are in the data.

## Uncorrected sequential tests will flood the scientific literature with false positives

We have demonstrated a possible problem with sequential tests on simulations. These results show that sequential correction strategies are more liberal than their simultaneous counterparts. Therefore we should expect more false positives if sequential correction methods were performed on a dataset. We now turn our attention to empirical data from a well-known shared dataset in neuroscience to examine the effect of multiple reuses of the dataset. This empirical example is to confirm the simulations and show that more positive findings (i.e. null hypothesis rejected) will be identified with sequential correction. We used 68 cortical thickness estimates from the 1200 subject release of the HCP dataset (*Van Essen et al., 2012*). All subjects belonging to this dataset gave informed consent (see *Van Essen et al., 2013* for more details). IRB protocol #31848 approved by the Stanford IRB approves the analysis of shared data. We then used 182 behavioral measures ranging from task performance to survey responses (see *Supplementary file 1*). For simplicity, we ignore all previous publications using the HCP dataset (of which there are now several hundred) for our p-value correction calculation.

We fit 182 linear models in which each behavior (dependent variable) was modeled as a function of each of the 68 cortical thickness estimates (independent variables), resulting in a total of 12,376 statistical tests. As a baseline, we corrected all statistical tests simultaneously with Bonferroni and FDR. For all other procedures, the independent variables within each mode (i.e.

cortical thickness) had simultaneous FDR correction while considering each linear model (i.e. each behavior) sequentially. The procedures considered were: uncorrected sequential analysis with both Bonferroni and FDR simultaneous correction procedures; all three sequential correction procedures with FDR simultaneous correction within each model. For the sequential tests, the orders were randomized in two ways: (i) uniformly; (ii) weighting the earlier tests to be the significant findings found during the baseline conditions (see Methods). The latter considers how the methods perform if there is a higher chance that researchers test hypotheses that produce positive findings earlier in the analysis sequence rather than later. Sequential analyses had the order of tests randomized 100 times.

We asked two questions with these models. First, we identified the number of positive findings that would be reported for the different correction methods (a positive finding is considered to be when the null hypothesis is rejected at $p < 0.05$, two tail). Second, we asked how many additional scientific articles would be published claiming to have identified a positive result (i.e. a null hypothesis has been rejected) for the different correction methods. Importantly, in this evaluation of empirical data, we are not necessarily concerned with the number of true relationships with this analysis. Primarily, we consider the differences in the inferred statistical relationships when comparing the different sequential correction procedures to a baseline of the simultaneous correction procedures. These simultaneous procedures allow us to contrast the sequential approaches with current practices (Bonferroni, a conservative procedure, and FDR, a more liberal measure). Thus any procedure that is more stringent than the Bonferroni baseline will be too conservative (more type II errors). Any procedure that is less stringent than FDR will have an increased false discovery rate, implying more false positives (relative to the true positives). Note that, we are tackling only issues regarding correction procedures to multiple hypothesis tests; determining the truth of any particular outcome would require additional replication.

*Figure 3* shows the results for all correction procedures. Using sequentially uncorrected tests leads to an increase in positive findings (30/44 Bonferroni/FDR), compared to a baseline of 2 findings when correcting for all tests simultaneously (for both Bonferroni and FDR procedures). The sequentially uncorrected procedures would also result in 29/30 (Bonferroni/FDR)

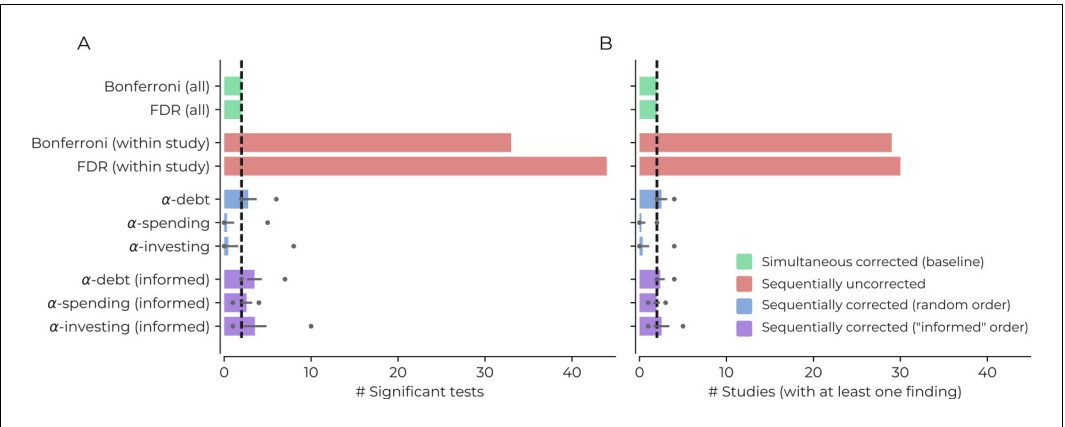

**Figure 3.** Demonstrating the impact of different correction procedures with a real dataset. (A) The number of significant statistical tests (x-axis) that are possible for various correction procedures in a real dataset from the Human Connectome Project: see the main text for more details, **supplementary file 1** for a list of the variables used in the analysis, and https://github.com/wiheto/datasetdecay copy archived at https://github.com/elifesciences-publications/datasetdecay for the code. (B) The potential number of publications (x-axis) that could result from the tests shown in panel A. This assumes that a publication requires a null hypothesis to be rejected in order to yield a positive finding. The dotted line shows the baseline from the two simultaneous correction procedures. Error bars show the standard deviation and circles mark min/max number of findings/studies for the sequential correction procedures with a randomly permuted test order.

publications that claim to identify at least one positive result instead of the simultaneous baseline of two publications (Bonferroni and FDR), reflecting a 1,400% increase in publications claiming positive results. If we accept that the two baseline estimates are a good trade-off between error rates, then we have good reason to believe this increase reflects false positives.

The sequential correction procedures were closer to baseline but saw divergence based on the order of the statistical tests. If the order was completely random, then $\alpha$-debt found, on average, 2.77 positive findings (min/max: 2/6) and 2.53 publications claiming positive results (min/max: 2/4) would be published. The random order leads to an increase in the number of false positives compared to baseline but considerably less than the sequentially uncorrected procedure. In contrast, $\alpha$-spending found 0.33 positive findings (min/max: 0/5) resulting in 0.22 studies with positive findings (min/max: 0/2) and $\alpha$-investing found 0.48 (min/max: 0/8) positive findings and 0.37 (min/max 0/4) studies with positive findings; both of which are below the conservative baseline of 2. When the order is informed by the baseline findings, the sequential corrections procedures increase in the number of findings (findings [min/max]: $\alpha$-debt: 3.49 [2/7], $\alpha$-spending: 2.58 [1/4], $\alpha$-investing: 3.54 [1/10]; and publications with positive findings [min/max]: $\alpha$-debt: 2.38 [2/4], $\alpha$-spending: 1.97 [1/3],

$\alpha$-investing: 2.54 [1/5]). All procedures now increase their number of findings above baseline. On average $\alpha$-debt with a random order has a 19% increase in the number of published studies with positive findings, substantially less than the increase in the number of uncorrected studies. Two conclusions emerge. First, $\alpha$-debt remains sensitive to the number of findings found regardless of the sequence of tests (fewer type II errors) and can never fall above the Bonferroni in regards to type II errors. At the same time, the other two sequential procedures can be more conservative than Bonferroni. Second, while $\alpha$-debt does not ensure the false positive rate remains under a specific level (more type I errors), it dramatically closes the gap between the uncorrected and simultaneous number of findings.

We have shown with both simulation and an empirical example of how sequential statistical tests, if left uncorrected, will lead to a rise of false positive results. Further, we have explored different sequential correction procedures and shown their susceptibility to both false negatives and false positives. Broadly, we conclude that the potential of a dataset to identify new statistically significant relationships will decay over time as the number of sequential statistical tests increases when controlling for sequential tests. In the rest of the discussion section, we first discuss the implications the different sequential

procedures have in regards to the desiderata outlined in the introduction. Then we discuss other possible solutions that could potentially mitigate dataset decay.

## Consequence for sequential tests and open data

We stated three desiderata for open data in the introduction: sharing incentive, open access, and a stable false positive rate. Having demonstrated some properties of sequential correction procedures, we revisit these aims and consider how the implementation of sequential correction procedures in practice would meet these desiderata. The current practice of leaving sequential hypothesis tests uncorrected leads to a dramatic increase in the false positive rate. While our proposed sequential correction techniques would mitigate this problem, all three require compromising on one or more of the desiderata (summarized in *Table 1*).

Implementing $\alpha$-spending would violate the sharing incentive desideratum as it forces the initial analysis to use a smaller statistical threshold to avoid using the entire wealth of $\alpha$. This change could potentially happen with appropriate institutional change, but placing restrictions on the initial investigator(s) (and increased type II error rate) would likely serve as a disincentive for those researchers to share their data. It also places incentives to restrict access to open data (violating the open access desideratum) as performing additional tests would lead to a more rapid decay in the ability to detect true positives in a given dataset.

Implementing $\alpha$-investing, would violate the open access desideratum for two reasons. First, like $\alpha$-spending there is an incentive to restrict incorrect statistical tests due to the sensitivity to order. Second, $\alpha$-investing would require tracking and time-stamping all statistical tests made on the dataset. Given the known issues of file drawer problem (*Rosenthal, 1979*), this is currently problematic to implement (see below). Also, publication bias for positive outcomes would result in statistical thresholds becoming more lenient over time with this correction procedure, which might lead to even more false positives (thus violating the no increase in false positives desideratum). Unless all statistical tests are time-stamped, which is possible but would require significant institutional change, this procedure would be hard to implement.

Implementing $\alpha$-debt would improve upon current practices but will compromise on the stable false positive rate desideratum. However, it will have little effect on the sharing incentive desideratum as the original study does not need to account for any future sequential tests. The open-access desideratum is also less likely to be compromised as it is less critical to identify the true-positives directly (i.e. it has the lowest type II error rate of the sequential procedures). Finally, while compromising the false positive desideratum, its false positive rate a marked improvement compared to sequentially uncorrected tests.

Finally, a practical issue that must be taken into consideration with all sequential correction procedures is whether it is ever possible to know the actual number of tests performed on an unrestricted dataset. This issue relates to the file drawer problem where there is a bias towards the publication of positive findings compared to null findings (*Rosenthal, 1979*). Until this is resolved, to fully sequentially correct for the number of previous tests corrected, an estimation of the number of tests may be required (e.g. by identifying publication biases; *Samartsidis et al., 2017*; *Simonsohn et al., 2013*). Using such estimations is less problematic with $\alpha$-debt because this only requires the number of tests to be known. Comparatively, $\alpha$-investing requires the entire results chain of statistical tests to be known and $\alpha$-spending requires knowing every $\alpha$ value that has been used, both of which would require additional assumptions to estimate. However, even if $\alpha$-debt correction underestimates the number of previous statistical tests, the number of false positives will be reduced compared to no sequential correction.

**Table 1.** Summary of the different sequential correction methods and the open-data desiderata. Yes indicates that the method is compatible with the desideratum.

|  | Sharing incentive | Open access | Stable false positive rate |
|---|---|---|---|
| $\alpha$-**spending** | No | No | Yes |
| $\alpha$-**investing** | Yes | No | Yes |
| $\alpha$-**debt** | Yes | Yes | No |

## Towards a solution

Statistics is a multifaceted tool for experimental researchers to use, but it (rarely) aims to provide universal solutions for all problems and use cases. Thus, it may be hard to expect a one size fits all solution to the problem of sequential tests on open data. Indeed, the idiosyncrasies within different disciplines regarding the size of data, open data infrastructure, and how often new data is collected, may necessitate that they adopt different solutions. Thus, any prescription we offer now is, at best, tentative. Further, the solutions also often compromise the desiderata in some way. That being said, there are some suggestions which should assist in mitigating the problem to different degrees. Some of these suggestions only require the individual researcher to adapt their practices, others require entire disciplines to form a consensus, and others require infrastructural changes. This section deals with solutions compatible with the null hypothesis testing framework, the next section considers solutions specific to other perspectives.

### Preregistration grace period of analyses prior to open data release

To increase the number of confirmatory analyses that can be performed on an open dataset, one possible solution is to have a 'preregistration grace period'. Here a description of the data can be provided, and data re-users will have the opportunity to write a preregistration prior to the data being released. This solution allows for confirmatory analyses to be performed on open data while simultaneously being part of different statistical families. This idea follows *Wagenmakers et al., 2012* definition of confirmatory analysis. Consequently, once the dataset or the first study using the dataset are published, the problems outlined in this paper will remain for all subsequent (non pre-registered) analyses reusing the data.

### Increased justification of the statistical family

One of the recurring problems regarding statistical testing is that, given the *Wagenmakers et al., 2012* definition, it is hard to class open data reuse as confirmatory after data release. However, if disciplines decide that confirmatory analyses on open data (post-publication) are possible, one of our main arguments above is that a new paper does not automatically create a new statistical family. If researchers

can, for other reasons, justify why their statistical family is separate in their analysis and state how it is different from previous statistical tests performed on the data, there is no necessity to sequentially correct. Thus providing sufficient justification for new a family in a paper can effectively reset the alpha wealth.

### Restrained or coordinated alpha-levels

One of the reasons the $\alpha$-values decays quickly in $\alpha$-invest and $\alpha$-spend is the 50% invest/spend rate that we chose in this article uses a large portion of the total $\alpha$-wealth in the initial statistical tests. For example, the first two tests in $\alpha$-spend, use 75% of the overall $\alpha$-wealth. Different spending or investing strategies are possible, which could restrain the decay of the remaining $\alpha$-wealth, allowing for more discoveries in later statistical tests. For example, a discipline could decide that the first ten statistical tests spend 5% of the $\alpha$ wealth, then the next ten spends 2.5% of the overall wealth. Such a strategy would still always remain under the overall wealth, but allow more people to utilize the dataset. However, imposing this restrained or fair-use of $\alpha$-spending would either require consensus from all researchers (however, this strategy would be in vain if just one researcher fails to comply) or restricting data access (compromising the open access desideratum). Importantly, this solution does not mitigate the decay of the alpha threshold; it just reduces the decay.

### Metadata about reuse coupled to datasets

One of the problems regarding sequential corrections is knowing how many tests have been made using the dataset. This issue was partially addressed above with suggestions for estimating the number of preceding tests. Additionally, repositories could provide information about all known previous uses of the data. Thus if data repositories were able to track summaries of tests performed and which variables involved in the tests, this would, at the very least, help guide future users with rough estimates. In order for this number to be precise, it would, however, require limiting the access to the dataset (compromising the open access desideratum).

### Held out data on repositories

A way to allow hypothesis testing or predictive frameworks (see below) to reuse the data is if the infrastructure exists that prevents the researcher from ever seeing some portion of the

data. Dataset repositories could hold out data which data re-users can query their results against to either replicate their findings or test their predictive models. This perspective has seen success in machine learning competitions which hold out test data. Additional requirements could be added to this perspective, such as requiring preregistrations in order to query the held out data. However, there have been concerns that held out data can lead to overfitting (e.g. by copying the best fitting model) (*Neto et al., 2016*) although others have argued this does not generally appear to be the case when evaluating overfitting (*Roelofs et al., 2019*). However, *Roelofs et al., 2019* noted that overfitting appears to occur on smaller datasets, which might prevent it from being a general solution for all disciplines.

### *Narrow hypotheses and minimal statistical families*

One way to avoid the sequential testing problem is to ensure small family sizes. If we can justify that there should be inherently small family sizes, then there is no need to worry about the sequential problems outlined here. This solution would also entail that each researcher does not need to justify their own particular family choice (as suggested above), but rather a specific consensus of what the contested concept *family* actually means is achieved. This would require: (1) confirmatory hypothesis testing on open data is possible, (2) encouraging narrow (i.e. very specific) hypotheses that will help maintain minimal family sizes, as the specificity of the hypothesis will limit the overlap with any other statistical test. Narrow hypotheses for confirmatory analyses can lead to families which are small, and can avoid correcting for multiple statistical tests (both simultaneous and sequential). This strategy is a possible solution to the problem. However, science does not merely consist of narrow hypotheses. Broader hypotheses can still be used in confirmatory studies (for example, genetic or neuroimaging datasets often ask broader questions not knowing which specific gene or brain area is involved, but know that a gene or brain region should be involved to confirm a hypothesis about a larger mechanism). Thus, while possibly solving a portion of the problem, this solution is unlikely to be a general solution for all fields, datasets, and types of hypotheses.

## Different perspective-specific solutions regarding sequential testing

The solutions above focused on possible solutions compatible within the null hypothesis testing framework to deal with sequential statistical tests, although many are compatible with other perspectives as well. There are a few other perspectives about data analysis and statistical inferences that are worth considering, three of which we discuss here. Each provide some perspective-specific solution to the sequential testing problem. Any of these possible avenues may be superior to the ones we have considered in this article, but none appear to readily applicable in all situations without some additional considerations.

The first alternative is Bayesian statistics. Multiple comparisons in Bayesian frameworks are often circumnavigated by partial pooling and regularizing priors (*Gelman et al., 2013*; *Kruschke and Liddell, 2017*). While Bayesian statistics can suffer from similar problems as NHST if misapplied (*Gigerenzer and Marewski, 2014*), it often deals with multiple tests without explicitly correcting for them, and may provide an avenue for sequential correction to be avoided. These techniques should allow for the sequential evaluation of different independent variables against a single dependent variable when using regularizing priors, especially as these different models could also be contrasted explicitly to see which model fits the data best. However, sequential tests could be problematic when the dependent variable changes and the false positive rate should be maintained across models. If uncorrected, this could create a similar sequential problem as outlined in the empirical example in the article. Nevertheless, there are multiple avenues where this could be fixed (e.g. sequentially adjusting the prior odds in Bayes-factor inferences). The extent of sequential analysis on open dataset within the Bayesian hypothesis testing frameworks, and possible solutions, is an avenue of future investigation.

The second alternative is using held-out data within prediction frameworks. Instead of using statistical inference, this framework evaluates a model by how well it performs on predicting unseen test data (*Yarkoni and Westfall, 2017*). However, a well-known problem when creating models to predict on test datasets is overfitting. This phenomenon occurs, for example, if a researcher queries the test dataset multiple times. Reusing test data will occur when

sequentially reusing open data. Held-out data on data repositories, as discussed above, is one potential solution here. Further, within machine learning, there have been advances towards having reusable held-out data that can be queried multiple times (*Dwork et al., 2015*; *Dwork et al., 2017*; *Rogers et al., 2019*). This avenue is promising, but there appear to be some drawbacks for sequential reuse. First, this line of work within 'adaptive data analysis' generally considers a single user querying the hold-out test data multiple times while optimizing their model/analysis. Second, this is ultimately a cross-validation technique which is not necessarily the best tool in datasets where sample sizes are small, (*Varoquaux, 2018*) which is often the case with open data and thus not a general solution to this problem. Third, additional assumptions exist in these methods (e.g., there is still a 'budget limit' in *Dwork et al., 2015*, and 'mostly guessing correctly' is required in *Rogers et al., 2019*). However, this avenue of research has the potential to provide a better solution than what we have proposed here.

The third and perhaps most radical alternative is to consider all open data analysis to be exploratory data analysis (EDA). In EDA, the primary utility becomes generating hypotheses and testing assumptions of methods (*Donoho, 2017*; *Jebb et al., 2017*; *Thompson et al., 2020*; *Tukey, 1980*). Some may still consider this reframing problematic, as it could make findings based on open data seem less important. However, accepting that all analyses on open data is EDA would involve less focus on statistical inference — the sequential testing problem disappears. An increase of EDA on exploratory analyses would lead to an increase of EDA results which may not replicate. However, this is not necessarily problematic. There would be no increase of false positives within *confirmatory studies* in the scientific literature and the increase EDA studies will provide a fruitful guide about which confirmatory studies to undertake. Implementing this suggestion would require little infrastructural or methodological change; however, it would require an institutional shift in how researchers interpret open data results. This suggestions of EDA on open data also fits with recent proposals calling for exploration to be conducted openly (*Thompson et al., 2020*).

## Conclusion
One of the benefits of open data is that it allows multiple perspectives to approach a question, given a particular sample. The trade-off of this benefit is that more false positives will enter the scientific literature. We remain strong advocates of open data and data sharing. We are not advocating that every single reuse of a dataset must necessarily correct for sequential tests and we have outlined multiple circumstances throughout this article where this is the case. However, researchers using openly shared data should be sensitive to the possibility of accumulating false positives and ensuing dataset decay that will occur with repeated reuse. Ensuring findings are replicated using independent samples will greatly decrease the false positive rate, since the chance of two identical false positives relationships occurring, even on well-explored datasets, is small.

## Methods

### Preliminary assumptions
In this article, we put forward the argument that sequential statistical tests on open data could lead to an increase in the number of false positives. This argument requires several assumptions regarding (1) the type of datasets analyzed; (2) what kind of statistical inferences are performed; (3) the types of sequential analyses considered.

### The type of dataset
we consider a dataset to be a fixed static snapshot of data collected at a specific time point. There are other cases of combining datasets or datasets that grow over time, but we will not consider those here. Second, we assume a dataset to be a random sample of a population and not a dataset that contains information about a full population.

### The type of statistical testing
We have framed our discussion of statistical inference using null hypothesis statistical testing (NHST). This assumption entails that we will use thresholded p-values to infer whether a finding differs from the null hypothesis. Our decision for this choice is motivated by a belief that the NHST framework being the most established framework for dealing with multiple statistical tests. There have been multiple valid critiques and suggestions to improve upon this statistical practice by moving away from thresholded p-values to evaluate hypotheses (*Cumming, 2014*; *Ioannidis, 2019*; *Lee, 2016*; *McShane et al., 2019*; *Wasserstein et al.,*

*2019*). Crucially, however, many proposed alternative approaches within statistical inference do not circumnavigate the problem of multiple statistical testing. For example, if confidence intervals are reported and used for inference regarding hypotheses, these should also be adjusted for multiple statistical tests (see, e.g. *Tukey, 1991*). Thus, any alternative statistical frameworks that still must correct for multiple simultaneous statistical testing will have the same sequential statistical testing problem that we outline here. Thus, while we have chosen NHST for simplicity and familiarity, this does not entail that the problem is isolated to NHST. Solutions for different frameworks may however differ (see the discussion section for Bayesian approaches and prediction-based inference perspectives).

The types of analyses

Sequential analyses involve statistical tests on the same data. Here, we consider sequential analyses that reuse the same data and analyses to be a part of the same statistical family (see section on statistical families for more details). Briefly, this involves either the statistical inferences being classed as exploratory or answering the same confirmatory hypothesis or research question. Further, we only consider analyses that are not supersets of previous analyses. This assumption entails that we are excluding analyses where a statistical model may improve upon a previous statistical model by, for example, adding an additional layer in a hierarchical model. Other types of data reuse may not be appropriate for sequential correction methods and are not considered here.

While we have restrained our analysis with these assumptions and definitions, it is done primarily to simplify the argument regarding the problem we are identifying. The degree to which sequential tests are problematic in more advanced cases remains outside the scope of this paper.

*Simulations*

The first simulation sampled data for one dependent variable and 100 independent variables from a multivariate Gaussian distribution (mean: 0, standard deviation: 1, covariance: 0). We conducted 100 different pairwise sequential analyses in a random order. For each analysis, we quantified the relationship between an independent and the dependent variable using a Pearson correlation. If the correlation had a two-tailed p-value less than 0.05, we considered it to

be a false positive. The simulation was repeated for 1000 iterations.

The second simulation had three additional variables. First, a variable that controlled the number of true positives in the data. This variable varied between 1-10. Second, the selected true positive variables, along with the dependent variable, had their covariance assigned as $p$. $p$ varied between 0 and 1 in steps of 0.025. Finally, we wanted to test the effect of the analysis order to identify when the true positive were included in the statistical tests. Each sequential analysis, ($m_1$, $m_2$, $m_3$ ...), could be assigned to be a 'true positive' (i.e., covariance of $p$ with the dependent variable) or a 'true negative' (covariance of 0 with dependent variable). First, $m_1$ would be assigned one of the trials, then $m_2$ and so forth. This procedure continued until there were only true positives or true negatives remaining. The procedure assigns the ith analysis to be randomly assigned, weighted by $\lambda$. If $\lambda$ was 0, then there was a 50% chance that $m_i$ would be a true positive or true negative. If $\lambda$ was 1, a true positive was 100% more likely to be assigned to $m_i$ (i.e. an odds ratio of 1+$\lambda$:1), The reverse occurred if $\lambda$ was negative (i.e. -1 meant a true negative was 100% more likely at $m_i$).

*Empirical example*

Data from the Human Connectome Project (HCP) 1200 subject release was used (*Van Essen et al., 2012*). We selected 68 estimates of cortical thickness to be the independent variables for 182 continuous behavioral and psychological variables dependent variables. Whenever possible, the age-adjusted values were used. *Supplementary file 1* shows the variables selected in the analysis.

For each analysis, we fit an ordinary least squares model using *Statsmodels* (0.10.0-dev0 +1579, https://github.com/statsmodels/statsmodels/). For all statistical models, we first standardized all variables to have a mean of 0 and a standard deviation of 1. We dropped any missing values for a subject for that specific analysis. Significance was considered for any independent variable if it had a p-value < 0.05, two-tailed for the different correction methods.

We then quantified the number of findings and the number of potential published studies with positive results that the different correction methods would present. The number of findings is the sum of independent variables that were considered positive findings (i.e. p < 0.05, two-tailed). The number of potential studies that

identify positive results is the number of dependent variables that had at least one positive finding. The rationale for the second metric is to consider how many potential non null-finding publications would exist in the literature if a separate group conducted each analysis.

For the sequential correction procedures, we used two different orderings of the tests. The first was with a uniformly random order. The second was an informed order that pretends we somehow a priori knew which variables will be correlated. The motivation behind an informed order is because it may be unrealistic that scientists ask sequential questions in a random order. The 'informed' order was created by identifying the significant statistical tests when using simultaneous correction procedures (see below). With the baseline results, we identified analyses which were *baseline positives* (i.e. significant with any of the simultaneous baseline procedures. There were two analyses) and the other analyses that were *baseline negatives*. Then, as in simulation 2, the first analysis, $m_1$ was randomly assigned to be a baseline positive or negative with equal probability. This informed ordering means that the baseline positives would usually appear in an earlier in the sequence order. All sequential correction procedures were applied 100 times.

### Simultaneous correction procedures

We used the Bonferroni method and the Benjamini and Hochberg FDR method for simultaneous correction procedures (*Benjamini and Hochberg, 1995*). Both correction methods were run using multipy (v0.16, https://github.com/puolival/multipy). The FDR correction procedure intends to limit the proportion of type I errors by keeping in below a certain level. In contrast, Bonferroni error intends to limit the probability of at least one type-I error. Despite ideological criticisms and objections to both these methods (Bonferroni: *Perneger, 1998*; FDR: *Mayo, 2017*), the Bonferroni correction is a conservative procedure that allows for more type II errors to occur and the FDR is a liberal method (i.e. allows for more type I errors). Together they offer a baseline range that allows us to contrast how the sequential correction procedures perform together.

In the second simulation, the false discovery rate was also calculated to evaluate different correction methods. To calculate this metric, the average number of true positives was divided by the average number of discoveries (average false positives + average true positives).

### Sequential correction procedures

*Uncorrected.* This procedure is to not correct for any sequential analyses. This analogous to reusing open data with no consideration for any sequential tests that occur due to data reuse. For all sequential hypothesis tests, p<0.05 was considered a significant or positive finding.

*α-debt.* A sequential correction procedure that, to our knowledge, has not previously been proposed. At the first hypothesis tested, $\alpha_1$ sets the statistical significance threshold (here 0.05). At the ith hypothesis tested the statistical threshold is $\alpha_i = \frac{\alpha_1}{i}$. The rationale here is that, at the ith test, a Bonferroni correction is applied that considers there to be $i$ number of tests performed. This method lets the false positive rate increase (i.e. the debt of reusing the dataset) as each test corrects for the overall number of tests, but all earlier tests have a more liberal threshold. The total possible 'debt' incurred for $m$ number of sequential tests can be calculated by $\sum_{i=1}^{m} \alpha_i$ and will determine the actual false positive rate.

*α-spending.* A predefined $\alpha_0$ is selected, which is called the $\alpha$-wealth. At the ith test the statistical threshold, $\alpha_i$, a value is selected to meet the condition that $\sum_{j=1}^{i} \alpha_j < \alpha_0$. The ith test selects $\alpha_i$ that spends part of the remaining 'α-wealth'. The remaining $\alpha$-wealth at test $i$ is $\alpha_0 - \sum_{j=1}^{i-1} \alpha_j$. Like, $\alpha$-debt, this method effectively decreases the p-value threshold of statistical significance at each test. However, it can also ensure that the false positive rate of all statistical tests is never higher than $\alpha_0$. Here, at test $i$ we always spend 50% of $\alpha_{i-1}$ and $\alpha_0$ is set to 0.05. See *Foster and Stine, 2008* for more details.

*α-investing.* The two previous methods only allow for the statistical threshold to decrease over time and are more akin to familywise error correction procedures. An alternative approach, which is closer to false discovery rate procedures, is to ensure the false discovery rate remains below a predefined wealth value ($W_0$) (*Foster and Stine, 2008*). At each test, $\alpha_i$ is selected from the remaining wealth at $W_{i-1}$. If the sequentially indexed test $i$ was considered statistically significant (i.e. rejecting the null hypothesis), then $W_i$ increases: $W_i = W_{i-1} + \omega$. Alternatively, if the null hypothesis cannot be rejected at $i$, then the wealth decreases: $W_i = W_{i-1} - \frac{\alpha_i}{1-\alpha_i}$. We set $\omega$ to $\alpha_0$, which is the convention, $\alpha_0$ to 0.05, and $\alpha_i$ is set to 50% of the remaining wealth. See *Foster and Stine, 2008* for more details.

When combining the simultaneous and sequential correction procedures in the empirical example, we used the sequential correction procedure to derive $\alpha_i$, which we then used as the threshold in the simultaneous correction.

## Acknowledgements
We thank Pontus Plavén-Sigray, Lieke de Boer, Nina Becker, Granville Matheson, Björn Schiffler, and Gitanjali Bhattacharjee for helpful discussions and feedback.

**William Hedley Thompson** is in the Department of Psychology, Stanford University, Stanford, United States, and the Department of Clinical Neuroscience, Karolinska Institutet, Stockholm, Sweden
william.thompson@stanford.edu
https://orcid.org/0000-0002-0533-6035

**Jessey Wright** is in the Department of Psychology and the Department of Philosophy, Stanford University, Stanford, United States
https://orcid.org/0000-0001-5003-0572

**Patrick G Bissett** is in the Department of Psychology, Stanford University, Stanford, United States

**Russell A Poldrack** is in the Department of Psychology, Stanford University, Stanford, United States
https://orcid.org/0000-0001-6755-0259

*Author contributions:* William Hedley Thompson, Conceptualization, Formal analysis, Investigation, Visualization, Methodology; Jessey Wright, Patrick G Bissett, Conceptualization; Russell A Poldrack, Conceptualization, Resources, Supervision

*Competing interests:* The authors declare that no competing interests exist.

### Funding

| Funder | Grant reference number | Author |
| --- | --- | --- |
| Knut och Alice Wallenbergs Stiftelse | 2016.0473 | William Hedley Thompson |

The funders had no role in study design, data collection and interpretation, or the decision to submit the work for publication.

### Decision letter and Author response
Decision letter https://doi.org/10.7554/eLife.53498.sa1
Author response https://doi.org/10.7554/eLife.53498.sa2

## Additional files

### Supplementary files
• Supplementary file 1. The variables selected for analysis in *Figure 3*.

• Transparent reporting form

### Data availability
All empirical data used in Figure 3 originates from the Human Connectome Project (https://www.humanconnectome.org/) from the 1200 healthy subject release. Code for the simulations and analyses is available at https://github.com/wiheto/datasetdecay.

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
