## [Decision Letter]

Thank you for submitting your article "Dataset decay and the problem of sequential analyses on open datasets" for consideration by *eLife*. Please note that following a discussion among the relevant editors, your article was considered as a Feature Article rather than as a Research Article.

Your article has been reviewed by three peer reviewers, and the evaluation has been overseen by a Reviewing Editor (Chris I Baker) and the *eLife* Features Editor (Peter Rodgers). The following individuals involved in review of your submission have agreed to reveal their identity: Chris I Baker (Reviewer #1); Nick Holmes (Reviewer #2); Guillaume A Rousselet (Reviewer #3).

Summary:

The reviewers all agreed that the manuscript focused on an important topic, and they all appreciated the simulations and analyses included it. However, the manuscript would benefit from clarifying a number of points - please see below. In particular, some passages require more in-depth discussion, and the passage that discuss potential solutions need to be fleshed out.

Essential revisions:

1) The notion of exploratory versus confirmatory analyses is ultimately a key issue in this manuscript. Indeed the authors propose that one solution to the problem of sequential analyses is to treat all studies using open data as exploratory. However, the authors do not clearly define or discuss these terms or the implications of labelling analyses as one or the other. I think the manuscript would benefit from more explicitly describing and discussing the distinction between exploratory and confirmatory, rather than assuming that everyone is already on the same page.

2) Another important issue is the ability to determine how many prior tests have been performed on a dataset. As the authors note several times in the discussion, the "file drawer" problem is of major concern here. But the authors do not really consider or discuss how this problem could possibly be dealt with. For example, should authors be required to preregister the analyses they plan to perform before being given access to a dataset? I think this is such an important issue in the context of the current manuscript that it deserves more in depth discussion - even if the field decides on an appropriate method of correction, that will only prove useful if there is a way to track the number of tests performed and not just those that resulted in a significant effect and hence publication.

3) In general, while the manuscript does a good job of highlighting the problem of sequential analyses on open datasets and discusses some possible solutions, it does not really suggest any recommendations for the path forward. How should the field grapple with these issues? Which of the possible solutions should be favored, if any? How should the field decide what is the best solution? How should we keep track of analyses on open datasets.

4) In the abstract, the authors state that "we should expect a dataset's utility for discovering new true relations between variables to decay'. I don't quite follow this. The alpha level is about controlling the false-positive rates. I do not see a clear link between this and the likelihood of new true-positive discoveries (which would require consideration of likely effect sizes, power, etc). If a researcher has a (true) hypothesis which is clearly-predicted by theory, convergent with other datasets, supported by a small experiment, and comes with a precise effect-size, I do not see why *any* number of prior tests of an open dataset should affect that open dataset's ability to support the researcher's well-defined hypothesis. These researchers could, indeed, simply abandon the null-hypothesis significance-testing approach for the open dataset, and simply ask: what are the likely values for the effect that I am quite sure exists? Perhaps by 'new true relations' the authors here mean 'unpredicted' or 'novel' or even 'random' and only within the NHST approach? So, my general comment here is that I am uncomfortable with the idea that open data sets decay in usefulness with time, and I would ask the authors to consider this, and soften or qualify the description.

5) I can see that the definition of what is a 'family' of tests is fundamental here. What I find a bit odd about this sequential testing approach is that, at the outset, the size of the family is not known. But it could be predicted based on prior or concurrent datasets, or could be set at an arbitrary level. Have the authors considered, for example: the first X hypothesis tests can use a certain alpha level, then the next X tests, etc. This stratified sequential approach would set the size of each family from the outset, and allow everyone in the first X groups of researchers to work on a level playing field (there would then, hopefully, be no great rush to be the first researchers to test their dubious hypotheses without thought, thus wasting their scoop of alpha).

6) The sequential correction methods all punish late-comers to the data party. Perhaps a particular dataset is perfect to test an idea which has not yet been developed - the data comes before the idea's time. It seems wrong that good researchers or good ideas who happen to arrive at the dataset late relative to other (worse) researchers or ideas should be 'punished' with higher alphas just for being late. (Not wishing to increase the admin-burden, ) perhaps some of the alpha can be saved up for a rainy day? Perhaps some of the alpha can be won or awarded through a competitive merit-based process? Perhaps researchers who meet a certain level of good research practice (e.g., pre-registration, ethical, open, all the necessary review, meta-analysis, and experimental work already in place, etc), should be allowed to use standard alpha levels, and it is only the disorganised vultures feeding on the data carcass who should be discouraged with alpha-punishment?

7) If the dataset comprises *all* the available data on a particular topic (e.g., brains affected by 'mad cow disease' in Britain in the late 1990s) - i.e., it is the population, and not just a sample - does this change the assumptions or outcomes of the authors' approach at all? It feels like it should be a special case, but maybe not...

8) Relatedly, if a dataset is large, one solution could be simply to restrict researchers to a random sample of the dataset (say, 10% of the available data), and allow them to keep alpha at standard levels. Because exactly the same data will almost never be tested twice on the same or a different hypothesis, does this completely remove the problems of inflated false-positives? It feels to me like it should. Should alpha correction only apply to researchers who use exactly the same subset and/or all the dataset?

9) In the authors' simulations, to estimate the likely number of publications resulting from false-positive findings, they assume that *every single 'significant' finding* will lead directly to a single publication - in effect, that publication bias is absolute, complete, and is itself unbiased. I find this assumption very hard to stomach. Researchers may tend to hold back significant results which don't support their, or their supervisors' or group's prior, current, or proposed research. Publication bias is not simply the immediate publication of (false) positive results, but also the delayed or suppressed publication of (true) negative or opposite-direction (false positive) results. Further, many (good) labs would replicate a result before going to press, or at least report multiple results, true and false, in the same paper. The authors may have stated this in other ways, but I think this strong assumption leads only to a very upper bound on the likely number of resulting (false positive) papers. Perhaps this can be stressed more clearly?

10) The authors used a real dataset to test a series of psychological hypotheses. They seem to have assumed that none of these hypotheses would pick up on any real effects in the data. Can they comment on the likelihood that their tests are establishing the true null distribution of effects, rather than actually being skewed by real effects? One solution would be to scramble the raw data in some way to ensure that even if there was a true effect, randomised and then processed voxels would not show this effect.

11) The introduction and discussion could do a better job at contrasting different empirical traditions and statistical approaches. The introduction could make clearer that the current project assumes that most researchers are engaged in a particular (though dominant) type of research involving confirmatory hypothesis testing in which the goal is to explain rather than predict. Do you think the problem would be different if the focus was on prediction?

The discussion mentions cross-validation and the problem with small sample sizes, but doesn't acknowledge explicitly the tension between explanatory and predictive research - Yarkoni & Westfall (2017) is a great reference on that topic:

https://doi.org/10.1177/1745691617693393

12) FDR is not clearly defined and would need to be better justified given the strong limitations of such a concept, which Richard Morey and Deborah Mayo described as completely flawed:

https://osf.io/ps38b/

https://medium.com/@richarddmorey/redefining-statistical-significance-the-statistical-arguments-ae9007bc1f91

13) What if we are in a field in which inappropriate statistical methods are the norm: should future researcher using appropriate tools be penalised for analysing a dataset after many doomed attempts? You touch indirectly on the subject in the section "Gray-area when families are not clear". For instance, in a field dominated by fixed effect analyses of means, I would argue that researchers attempting to fit carefully justified generalised hierarchical models should be allowed to reset their alpha counter.

14) The discussion mentions Bayesian statistics as a potential solution, but with the current trend in adopting arbitrary thresholds for Bayes factors, the same problems encountered in mindless frequentist practices will also apply to Bayesian/Laplacian ones:

Gigerenzer, G. & Marewski, J.N. (2015) Surrogate Science: The Idol of a Universal Method for Scientific Inference. Journal of Management, 41, 421-440.

---

## [Author Response]

The reviewers comments were very helpful. We have three new subsections to the manuscript. One at the start, one in the methods, and one new section in the discussion. These address points raised by the reviewers, so we have decided to provide them in full here and motivate them generally first, before addressing each Essential revision point individually.

# General response 1. New section: “Preliminary assumptions” (Methods section)

First, the reviewers have raised, on multiple occasions, issues regarding NHST, proposed alternatives to p-values, types of datasets, and asked about certain types of inferences that could be made. Thus, we have written an “assumptions” section to motivate certain choices in our analysis and discuss the implications of these choices. This section reads:

“Preliminary assumptions

In this article, we put forward the argument that sequential statistical tests on open data could lead to an increase in the number of false positives. This argument requires several assumptions regarding (1) the type of datasets analysed; (2) what kind of statistical inferences are performed; (3) the types of sequential analyses considered.

*The type of dataset:* we consider a dataset to be a fixed static snapshot of data collected at a specific time point. There are other cases of combining datasets or datasets that grow over time, but we will not consider those here. Second, we assume a dataset to be a random sample of a population and not a dataset that contains information about a full population.

*The type of statistical testing:* We have framed our discussion of statistical inference using null hypothesis statistical testing (NHST). This assumption entails that we will use thresholded p-values to infer whether a finding differs from the null hypothesis. Our decision for this choice is motivated by a belief that the NHST framework being the most established framework for dealing with multiple statistical tests. There have been multiple valid critiques and suggestions to improve upon this statistical practice by moving away from thresholded p-values to evaluate hypotheses (Cumming, 2014; Ioannidis, 2019; Lee, 2016; McShane, Gal, Gelman, Robert, & Tackett, 2019; Wasserstein, Schirm, & Lazar, 2019). Crucially, however, many proposed alternative approaches within statistical inference do not circumnavigate the problem of multiple statistical testing. For example, if confidence intervals are reported and used for inference regarding hypotheses, these should also be adjusted for multiple statistical tests (see, e.g. Tukey (1991)). Thus, any alternative statistical frameworks that still must correct for multiple simultaneous statistical testing will have the same sequential statistical testing problem that we outline here. Thus, while we have chosen NHST for simplicity and familiarity, this does not entail that the problem is isolated to NHST. Solutions for different frameworks may however differ (see the discussion section for Bayesian approaches and prediction-based inference perspectives).

*The types of analyses:* Sequential analyses involve statistical tests on the same data. Here, we consider sequential analyses that reuse the same data and analyses to be a part of the same statistical family (see section on statistical families for more details). Briefly, this involves either the statistical inferences being classed as exploratory or answering the same confirmatory hypothesis or research question. Further, we only consider analyses that are not supersets of previous analyses. This assumption entails that we are excluding analyses where a statistical model may improve upon a previous statistical model by, for example, adding an additional layer in a hierarchical model. Other types of data reuse may not be appropriate for sequential correction methods and are not considered here.

While we have restrained our analysis with these assumptions and definitions, it is done primarily to simplify the argument regarding the problem we are identifying. The degree to which sequential tests are problematic in more advanced cases remains outside the scope of this paper.”

We believe this section addresses many of the concerns raised by the reviewers. It clarifies our argument but does not reduce the importance of our results.

#General response 2. New section: “An intuitive example of the problem”

We feared that our argument was getting interpreted as a conceptual debate about what should be included in a “family” of tests where the simple solution is to say “we decided to have small statistical families”. While important (which we address later), we feared this was making the manuscript lose its focus. Thus, before the discussion of statistical families, the simulations, and empirical results, we present a simple example of the problem:

**“**An intuitive example of the problem

Before proceeding with technical details of the problem, we outline an intuitive problem regarding sequential statistical testing and open data. Imagine there is a dataset which contains the variables (𝑣1, 𝑣2, 𝑣3). Let us now imagine that one researcher performs the statistical tests to analyze the relationship between 𝑣1 ∼ 𝑣2 and 𝑣1 ∼ 𝑣3 and decides that a 𝑝<0.05 is treated as a positive finding (i.e. null hypothesis rejected). The analysis yields p-values of 𝑝=0.001 and 𝑝=0.04 respectively. In many cases, we expect the researcher to correct for the fact that two statistical tests are being performed. Thus, the researcher chooses to apply a Bonferroni correction such that p < 0.025 is the adjusted threshold for statistical significance. In this case, both tests are published, but only one of the findings is treated as a positive finding.

Alternatively, let us consider a different scenario with sequential analyses and open data. Instead, the researcher only performs one statistical test (𝑣1 ∼ 𝑣2, p=0.001). No correction is performed, and it is considered a positive finding (i.e. null hypothesis rejected). The dataset is then published online. A second researcher now performs the second test (𝑣1 ∼ 𝑣3, p=0.04) and deems this a positive finding too because it is under a 𝑝<0.05 threshold and they have only performed one statistical test. In this scenario, with the same data, we have two published positive findings compared to the single positive finding in the previous scenario. Unless a reasonable justification exists for this difference between the two scenarios, this is troubling.

What are the consequences of these two different scenarios? A famous example of the consequences of uncorrected multiple simultaneous statistical tests is the finding of fMRI BOLD activation in a dead salmon when appropriate corrections for multiple tests were not performed (Bennett, Baird, Miller, & George, 2010; Bennett, Wolford, & Miller, 2009). Now let us imagine this dead salmon dataset is published online but, in the original analysis, only one part of the salmon was analyzed, and no evidence was found supporting the hypothesis of neural activity in a dead salmon. Subsequent researchers could access this dataset, test different regions of the salmon and report their uncorrected findings. Eventually, we would see reports of dead salmon activations if no sequential correction strategy is applied, but each of these individual findings would appear completely legitimate by current correction standards.

We will now explore the idea of sequential tests in more detail, but this example highlights some crucial arguments that need to be discussed. Can we justify the sequential analysis without correcting for sequential tests? If not, what methods could sequentially correct for the multiple statistical tests? In order to fully grapple with these questions, we first need to discuss the notion of a statistical family and whether sequential analyses create new families.”

We hope the reviewers agree this section hopes makes the argument clearer by providing this example.

#*General response 3. New section: “Towards a solution”*

A considerable number of the suggestions made by the reviewers were regarding possible solutions to the problem. Many were good suggestions, but many do compromise one of the three desiderata we had outlined in the introduction (that the reviewers did not challenge) and one of our conclusions is that we will have to compromise on one of these desiderata. However, we agree that it is worthwhile spending some time discussing possible solutions. Thus, we have listed some of the possible solutions that could be implemented in the Discussion section:

“Towards a solution

Statistics is a multifaceted tool for experimental researchers to use, but it (rarely) aims to provide universal solutions for all problems and use cases. Thus, it may be hard to expect a one size fits all solution to the problem of sequential tests on open data. Indeed, the idiosyncrasies within different disciplines regarding the size of data, open data infrastructure, and how often new data is collected, may necessitate that they adopt different solutions. Thus, any prescription we offer now is, at best, tentative. Further, the solutions also often compromise the desiderata in some way. That being said, there are some suggestions which should assist in mitigating the problem to different degrees. Some of these suggestions only require the individual researcher to adapt their practices, others require entire disciplines to form a consensus, and others require infrastructural changes. This section deals with solutions compatible with the null hypothesis testing framework, the next section considers solutions specific to other perspectives.

*Preregistration grace period of analyses prior to open data release.* To increase the number of confirmatory analyses that can be performed on an open dataset, one possible solution is to have a "preregistration grace period". Here a description of the data can be provided, and data re-users will have the opportunity to write a preregistration prior to the data being released. This solution allows for confirmatory analyses to be performed on open data while simultaneously being part of different statistical families. This idea follows Wagenmakers et al. (2012) definition of confirmatory analysis. Consequently, once the dataset or the first study using the dataset are published, the problems outlined in this paper will remain for all subsequent (non pre-registered) analyses reusing the data.

*Increased justification of the statistical family.* One of the recurring problems regarding statistical testing is that, given the Wagenmakers et al. (2012) definition, it is hard to class open data reuse as confirmatory after data release. However, if disciplines decide that confirmatory analyses on open data (post-publication) are possible, one of our main arguments above is that a new paper does not automatically create a new statistical family. If researchers can, for other reasons, justify why their statistical family is separate in their analysis and state how it is different from previous statistical tests performed on the data, there is no necessity to sequentially correct. Thus providing sufficient justification for new a family in a paper can effectively reset the alpha wealth.

*Restrained or coordinated alpha-levels.* One of the reasons the $\ One of the reasons the $\alpha$-values decays quickly in $\alpha$-invest and $\alpha$-spend is the 50% invest/spend rate that we chose in this article uses a large portion of the total $alpha$ in the initial statistical tests. For example, the first two tests in $\alpha$-spend, use 75% of the overall $\alpha$-wealth. Different spending or investing strategies are possible, which could restrain the decay of the remaining $\alpha-wealth$, allowing for more discoveries in later statistical tests. For example, a discipline could decide that the first ten statistical tests spend 5% of the $\alpha$ wealth, then the next ten spends 2.5% of the overall wealth. Such a strategy would still always remain under the overall wealth, but allow more people to utilize the dataset. However, imposing this restrained or fair-use of $\alpha$ spending would either require consensus from all researchers (however, this strategy would be in vain if just one researcher fails to comply) or restricting data access (compromising the open access desideratum). Importantly, this solution does not mitigate the decay of the alpha threshold; it just reduces the decay.

One of the reasons the 𝛼-values decays quickly in 𝛼-invest and 𝛼-spend is the 50% invest/spend rate that we chose in this article uses a large portion of the total 𝑎𝑙𝑝ℎ𝑎 in the initial statistical tests. For example, the first two tests in 𝛼-spend, use 75% of the overall 𝛼-wealth. Different spending or investing strategies are possible, which could restrain the decay of the remaining 𝛼 − 𝑤𝑒𝑎𝑙𝑡ℎ, allowing for more discoveries in later statistical tests. For example, a discipline could decide that the first ten statistical tests spend 5% of the 𝛼 wealth, then the next ten spends 2.5% of the overall wealth. Such a strategy would still always remain under the overall wealth, but allow more people to utilize the dataset. However, imposing this restrained or fair-use of 𝛼 spending would either require consensus from all researchers (however, this strategy would be in vain if just one researcher fails to comply) or restricting data access (compromising the open access desideratum). Importantly, this solution does not mitigate the decay of the alpha threshold; it just reduces the decay.

*Metadata about reuse coupled to datasets.* One of the problems regarding sequential corrections is knowing how many tests have been made using the dataset. This issue was partially addressed above with suggestions for estimating the number of preceding tests. Additionally, repositories could provide information about all known previous uses of the data. Thus if data repositories were able to track summaries of tests performed and which variables involved in the tests, this would, at the very least, help guide future users with rough estimates. In order for this number to be precise, it would, however, require limiting the access to the dataset (compromising the open access desideratum).

qHeld out data on repositories. A way to allow hypothesis testing or predictive frameworks (see below) to reuse the data is if the infrastructure exists that prevents the researcher from ever seeing some portion of the data. Dataset repositories could hold out data which data re-users can query their results against to either replicate their findings or test their predictive models. This perspective has seen success in machine learning competitions which hold out test data. Additional requirements could be added to this perspective, such as requiring preregistrations in order to query the held out data. However, there have been concerns that held out data can lead to overfitting (e.g. by copying the best fitting model) (Neto et al., 2016) although others have argued this does not generally appear to be the case when evaluating overfitting (Roelofs et al., 2019). However, Roelofs et al. (2019) noted that overfitting appears to occur on smaller datasets, which might prevent it from being a general solution for all disciplines.

*Narrow hypotheses and minimal statistical families.* One way to avoid the sequential testing problem is to ensure small family sizes. If we can justify that there should be inherently small family sizes, then there is no need to worry about the sequential problems outlined here. This solution would also entail that each researcher does not need to justify their own particular family choice (as suggested above), but rather a specific consensus of what the contested concept family actually means is achieved. This would require: (1) confirmatory hypothesis testing on open data is possible, (2) encouraging narrow (i.e. very specific) hypotheses that will help maintain minimal family sizes, as the specificity of the hypothesis will limit the overlap with any other statistical test. Narrow hypotheses for confirmatory analyses can lead to families which are small, and can avoid correcting for multiple statistical tests (both simultaneous and sequential). This strategy is a possible solution to the problem. However, science does not merely consist of narrow hypotheses. Broader hypotheses can still be used in confirmatory studies (for example, genetic or neuroimaging datasets often ask broader questions not knowing which specific gene or brain area is involved, but know that a gene or brain region should be involved to confirm a hypothesis about a larger mechanism). Thus, while possibly solving a portion of the problem, this solution is unlikely to be a general solution for all fields, datasets, and types of hypotheses.”

[We repeat the reviewers’ points here in italic, followed by our reply and a description of the changes made].

Essential revisions:1) The notion of exploratory versus confirmatory analyses is ultimately a key issue in this manuscript. Indeed the authors propose that one solution to the problem of sequential analyses is to treat all studies using open data as exploratory. However, the authors do not clearly define or discuss these terms or the implications of labelling analyses as one or the other. I think the manuscript would benefit from more explicitly describing and discussing the distinction between exploratory and confirmatory, rather than assuming that everyone is already on the same page.

This is indeed a distinction that is lurking in the background of the text and we agree with the reviewers that this could be made explicit. We also had not explicitly highlighted the consequences of the more stringent definitions of confirmatory research and how this impacts statistical families (especially in sequential analyses). We have thus added two paragraphs to the section “Statistical families” [emphasis added on new text]:

“A family is a set of tests which we relate the same error rate to (familywise error). What constitutes a family has been challenging to precisely define, and the existing guidelines often contain additional imprecise terminology (e.g. Cox, 1965; Games, 1971; Hancock & Klockars, 1996; Hochberg & Tamhane, 1987; Miller, 1981). Generally, tests are considered part of a family when: (i) multiple variables are being tested with no predefined hypothesis (i.e. exploration or data-dredging), or (ii) multiple pre-specified tests together help support the same or associated research questions (Hancock & Klockars, 1996; Hochberg & Tamhane, 1987). Even if following these guidelines, there can still be considerable disagreements about what constituents a statistical family, which can include both very liberal and very conservative inclusion criteria. *An example of this discrepancy is seen in using a factorial ANOVA. Some have argued that the main effect and interaction are separate families as they answer 'conceptually distinct questions' (e.g. page 291 of Maxwell & Delaney, 2004), while others would argue the opposite and state they are the same family (e.g. Cramer et al., 2016; 136 Hancock & Klockars, 1996). Given the substantial leeway regarding the definition of family, recommendations have directed researchers to define and justify their family of tests a priori (Hancock & Klockars, 1996; Miller, 1981).*

A crucial distinction in the definition of a family is whether the analysis is confirmatory (i.e. hypothesis driven) or exploratory. Given issues regarding replication in recent years (Open Science Collaboration, 2015), there has been considerable effort placed into clearly demarcating what is exploratory and what is confirmatory. One prominent definition is that confirmatory research requires preregistration before seeing the data (Wagenmakers, Wetzels, Borsboom, Maas, & Kievit, 2012). However, current practice often involves releasing open data with the original research article. Thus, all data reuse may be guided by the original or subsequent analyses (a HARKing-like problem where methods are formulated after some results are known (Button, 2019)). Therefore, if adopting this prominent definition of confirmatory research (Wagenmakers et al., 2012), it follows that any reuse of open data after publication must be exploratory unless the analysis is preregistered before the data release.

Some may find Wagenmakers et al. (2012) definition to be too stringent and instead would rather allow that confirmatory hypotheses can be stated at later dates despite the researchers having some information about the data from previous use. Others have said confirmatory analyses may not require preregistrations (Jebb, Parrigon, & Woo, 2017) and have argued that confirmatory analyses on open data are possible (Weston et al., 2019). If analyses on open data can be considered confirmatory, then we need to consider the second guideline about whether statistical tests are answering similar or the same research questions. The answer to this question is not always obvious, as was highlighted above regarding factorial ANOVA. However, if a study reusing data can justify itself as confirmatory, then it must also justify that it is asking a 'conceptually distinct question' from previous instances that used the data. We are not claiming that this is not possible to justify, but the justification ought to be done if no sequential correction is applied as new families are not created just because the data is being reused (see next section).”

We believe that all this new text to help clarify what statistical families are made the previous supplementary information that tried to give examples has become less useful. Thus, it has been removed.

2) Another important issue is the ability to determine how many prior tests have been performed on a dataset. As the authors note several times in the discussion, the "file drawer" problem is of major concern here. But the authors do not really consider or discuss how this problem could possibly be dealt with. For example, should authors be required to preregister the analyses they plan to perform before being given access to a dataset? I think this is such an important issue in the context of the current manuscript that it deserves more in depth discussion - even if the field decides on an appropriate method of correction, that will only prove useful if there is a way to track the number of tests performed and not just those that resulted in a significant effect and hence publication.

The file drawer problem is indeed a problem that is lurking in the background. We have made two different changes to address this. Firstly we have edited the manuscript substantially to remove the implication that only statistically significant findings are publishable. Instead we now talk about studies that have either positive findings or null findings, to prevent us from unintentionally promoting file drawer scenarios or behaviour.

Second, in relation to the file drawer problem, it is also relevant with regards to estimating the number of sequential tests that have been performed. We discuss the final paragraph of the subsection “Consequence for sequential tests and open data” in the Discussion:

“Finally, a practical issue that must be taken into consideration with all sequential correction procedures is whether it is ever possible to know the actual number of tests performed on an unrestricted dataset. This issue relates to the file drawer problem where there is a bias towards the publication of positive findings compared to null findings (Rosenthal, 1979). Until this is resolved, to fully sequentially correct for the number of previous tests corrected, an estimation of the number of tests may be required (e.g. by identifying publication biases (Samartsidis et al., 2017; Simonsohn, Nelson, & Simmons, 2013)). Using such estimations is less problematic with 𝛼-debt because this only requires the number of tests to be known. Comparatively, 𝛼-investing requires the entire results chain of statistical tests to be known and 𝛼-spending requires knowing every 𝛼 value that has been used, both of which would require additional assumptions to estimate. However, even if 𝛼-debt correction underestimates the number of previous statistical tests, the number of false positives will be reduced compared to no sequential correction.”

3) In general, while the manuscript does a good job of highlighting the problem of sequential analyses on open datasets and discusses some possible solutions, it does not really suggest any recommendations for the path forward. How should the field grapple with these issues? Which of the possible solutions should be favored, if any? How should the field decide what is the best solution? How should we keep track of analyses on open datasets.

We have now included an entire section about possible solutions going forward. See General response 3 above for the text that we have added. It also discusses how we will arrive at the different solutions (i.e. through discipline consensus or infrastructural changes).

4) In the abstract, the authors state that "we should expect a dataset's utility for discovering new true relations between variables to decay'. I don't quite follow this. The alpha level is about controlling the false-positive rates. I do not see a clear link between this and the likelihood of new true-positive discoveries (which would require consideration of likely effect sizes, power, etc). If a researcher has a (true) hypothesis which is clearly-predicted by theory, convergent with other datasets, supported by a small experiment, and comes with a precise effect-size, I do not see why *any* number of prior tests of an open dataset should affect that open dataset's ability to support the researcher's well-defined hypothesis. These researchers could, indeed, simply abandon the null-hypothesis significance-testing approach for the open dataset, and simply ask: what are the likely values for the effect that I am quite sure exists? Perhaps by 'new true relations' the authors here mean 'unpredicted' or 'novel' or even 'random' and only within the NHST approach? So, my general comment here is that I am uncomfortable with the idea that open data sets decay in usefulness with time, and I would ask the authors to consider this, and soften or qualify the description.

The logic of our argument is as follows: *If* we decide to control for the increasingly likelihood of false positives, *then* it will become harder to identify true positives. This appears true for all types of hypothesis testing, not just NHST (see General response 1). Obviously, if there is no wish to control for the increase in false positives, then the ability to identify true positives becomes unimpeded. Thus for increased precision in the text, we have amended the Abstract to say (emphasis added to show additions):

"*Thus, if correcting for this increase in hypothesis testing,* we should expect a dataset’s utility for discovering new true relations between variables to decay."

And have changed the Discussion to say (emphasis added to show additions):

"Broadly, we conclude that a dataset’s potential to identify new statistically significant relationships will decay over time as the number of sequential statistical tests increases *when controlling for sequential tests."*

These additions qualify that the “decay” for true positives that we discuss is contingent on controlling for the increase in false positives.

5) I can see that the definition of what is a 'family' of tests is fundamental here. What I find a bit odd about this sequential testing approach is that, at the outset, the size of the family is not known. But it could be predicted based on prior or concurrent datasets, or could be set at an arbitrary level. Have the authors considered, for example: the first X hypothesis tests can use a certain alpha level, then the next X tests, etc. This stratified sequential approach would set the size of each family from the outset, and allow everyone in the first X groups of researchers to work on a level playing field (there would then, hopefully, be no great rush to be the first researchers to test their dubious hypotheses without thought, thus wasting their scoop of alpha).

Firstly, see the reply to Essential revision 1 about changes made to the section on statistical families. See also General response 2 for a intuitive problem about sequential testing as this may help assist why it is a problem despite the size of the family being unknown. Also, in General response 3, we have included this stratified approach as a possible solution. However, we note that this does not stop the decay of alpha, it just reduces the decay by starting lower.

To address this comment in a little more detail, the investing and spending correction procedures allow for infinite size families (as alpha will just get infinitesimally small, and never exceeds the set amount).

It is indeed possible for the reviewer’s approach to be an alternative way of doing sequential correction. In alpha-spending (the simplest of the correction procedures) we, throughout the paper, spent 50% of the remaining alpha-wealth. We could have changed the spending rate (e.g. only spend 10%) or varied the spend rate but this has little effect on the conclusions of the paper. It is indeed possible for different spending procedures to occur. Thus it is possible for the first ten to spend 5% of the overall alpha-wealth (coming to 50%). Then the next ten researchers spend 50% of the remaining wealth (coming to 75%). In practice this would mean alpha threshold (assuming 0.05 is the wealth of): 0.0025 for the first ten tests, 0.0013 for the next ten tests. This strategy will however always have a lower alpha than the alpha-debt approach and does not change any of the conclusions of the paper. So changing the spending or investing rates will change the thresholds, it does not get around the problem – it would just reduce the rate of decay.

6) The sequential correction methods all punish late-comers to the data party. Perhaps a particular dataset is perfect to test an idea which has not yet been developed - the data comes before the idea's time. It seems wrong that good researchers or good ideas who happen to arrive at the dataset late relative to other (worse) researchers or ideas should be 'punished' with higher alphas just for being late. (Not wishing to increase the admin-burden, ) perhaps some of the alpha can be saved up for a rainy day? Perhaps some of the alpha can be won or awarded through a competitive merit-based process? Perhaps researchers who meet a certain level of good research practice (e.g., pre-registration, ethical, open, all the necessary review, meta-analysis, and experimental work already in place, etc), should be allowed to use standard alpha levels, and it is only the disorganised vultures feeding on the data carcass who should be discouraged with alpha-punishment?

The idea behind “dataset decay” is that it will ultimately punish late comers *if* they are considered the same type of statistical family. General response 3 discusses some possible solutions more forward that incorporates some of the suggestions the reviewer has here. However, as we have discussed in Essential revision 1, we have further clarified our discussion of statistical families in order to make it clear that it is possible to have “standard alpha levels” if (1) the analysis is classed as confirmatory, and (2) that any data reuser can justify that their new analysis is a new hypothesis from previous use-cases.

7) If the dataset comprises *all* the available data on a particular topic (e.g., brains affected by 'mad cow disease' in Britain in the late 1990s) - i.e., it is the population, and not just a sample - does this change the assumptions or outcomes of the authors' approach at all? It feels like it should be a special case, but maybe not…

This is an excellent point. However, our intention in this article is to raise the issue: there is a subset of analyses that can be conducted on open data which could increase the number of false positives. It is not our intention to say that all analyses on open data are problematic. We have tried to explicitly define the types of analyses and datasets we are considered in General response 1 (preliminary assumptions). This should hopefully not leave a reader confused about what type of datasets we are talking about (the types of analyses where one has access to the full dataset means that there is no longer the same type of uncertainty as when taking a random sample). The new section in General response 1 explicitly says that we are not considering circumstances when the dataset is a whole population to avoid this confusion.

We have also raised in General response 3 (towards a solution) the point that different fields may require different solutions to the sequential problem depending on the type of datasets they have. Thus, with this disclaimer, we hope the reviewers agree that the reader understands which type of datasets we are addressing.

8) Relatedly, if a dataset is large, one solution could be simply to restrict researchers to a random sample of the dataset (say, 10% of the available data), and allow them to keep alpha at standard levels. Because exactly the same data will almost never be tested twice on the same or a different hypothesis, does this completely remove the problems of inflated false-positives? It feels to me like it should. Should alpha correction only apply to researchers who use exactly the same subset and/or all the dataset?

We do not see this as a potential solution, unfortunately. We had already discussed the reuseable hold out approach as a possible solution. The reviewer here is advocating for each data-user to create their out hold out data. This approach is still not ideal because any knowledge gained about the dataset in the previous analyses could bias the methods (see Essential revision 1). This will ultimately lead to overfitting as, results from previous uses of the data will guide the analyses. Thus, any knowledge about previous uses of the data means that there is knowledge about the test dataset, which increases the chance of overestimating the prediction. This is a well documented problem in machine learning.

Further, this approach requires for datasets to be sufficiently powered to achieve not just apt training models but sufficient data to assess predictive generalizability (we already touch upon this in our discussion about the reuseable hold out dataset), which may not be a universal solution for sequential tests on open data.

When we discuss the reuseable held out data, we are now more explicit that this is within a predictive framework. Further in the new subsection “Towards a solution” (General response 3) we discuss the possibilities of having a collective held out dataset on repositories, which does not seem to lead to have led to extensive overfitting in machine learning yet. Indeed, this does seem like a good solution (although some still fear this will lead to overfitting) which will however require infrastructural change within many fields.

Finally, as discussed above, in the new section “Preliminary assumptions” (General response 1) we discuss that we are only treating analyses on complete datasets. We understand (and sympathize) with the reviewers wish for asking about whether the point still holds under certain assumptions (whole dataset or subset). We however feel the point of the paper is to establish the problem that can exist within many reuse cases (the simplest types), we do not think it should be our priority to show the extent of the problem or present a solution with regards to every possible type of statistical test and dataset type.

9) In the authors' simulations, to estimate the likely number of publications resulting from false-positive findings, they assume that *every single 'significant' finding* will lead directly to a single publication - in effect, that publication bias is absolute, complete, and is itself unbiased. I find this assumption very hard to stomach. Researchers may tend to hold back significant results which don't support their, or their supervisors' or group's prior, current, or proposed research. Publication bias is not simply the immediate publication of (false) positive results, but also the delayed or suppressed publication of (true) negative or opposite-direction (false positive) results. Further, many (good) labs would replicate a result before going to press, or at least report multiple results, true and false, in the same paper. The authors may have stated this in other ways, but I think this strong assumption leads only to a very upper bound on the likely number of resulting (false positive) papers. Perhaps this can be stressed more clearly?

The reviewers are correct that we may have been somewhat clumsy in our formulation. It was indeed unwise in our former formulation to present the results with a publication bias by discussing “number of publications”, where we should state the general consequences (with or without publication bias) by discussing the number of “positive findings”. Thus have rephrased the text in many places to state “number publications with positive findings (i.e. null hypothesis rejected)” instead of “number of publications”.

Several examples of the new formulation are listed below.

[In section: “Uncorrected sequential tests will flood the scientific literature with false positives”. Most of the text has been revised in someway, so no emphasis added.].

"We asked two questions with these models. First, we identified the number of positive findings that would be reported (a positive finding is considered to be when the null hypothesis is rejected at p < 0.05, two tail) for the different correction methods. Second, we asked how many additional scientific articles would be published claiming to have identified a positive result (i.e. a null hypothesis has been rejected) for the different correction methods. Importantly, in this evaluation of empirical data, we are not necessarily concerned with the number of true relationships with this analysis. Primarily, we consider the differences in the inferred statistical relationships when comparing the different sequential correction procedures to a baseline of the simultaneous correction procedures. These simultaneous procedures allow us to contrast the sequential approaches with current practices (Bonferroni, a conservative procedure, and FDR, a more liberal measure). Thus any procedure that is more stringent than the Bonferroni baseline will be too conservative (more type II errors). Any procedure that is less stringent than FDR will have an increased false discovery rate, implying more false positives (relative to the true positives). Note that, we are tackling only issues regarding correction procedures to multiple hypothesis tests; determining the truth of any particular outcome would require additional replication.

Figure 3 shows the results for all correction procedures. Using sequentially uncorrected tests leads to an increase in positive findings (30/44 Bonferroni/FDR), compared to a baseline of 2 findings when correcting for all tests simultaneously (for both Bonferroni and FDR procedures). The sequentially uncorrected procedures would also result in 29/30 (Bonferroni/FDR) publications that claim to identify at least one positive result instead of the simultaneous baseline of two publications (Bonferroni and FDR), reflecting a 1,400% increase in publications claiming positive results. If we accept that the two baseline estimates are a good trade-off between error rates, then we have good reason to believe this increase reflects false positives.

The sequential correction procedures were closer to baseline but saw divergence based on the order of the statistical tests. If the order was completely random, then 𝛼-debt found, on average, 2.77 positive findings (min/max: 2/6) and 2.53 publications claiming positive results (min/max: 2/4) would be published. The random order leads to an increase in the number of false positives compared to baseline but considerably less than the sequentially uncorrected procedure. In contrast, 𝛼-spending found 0.33 positive findings (min/max: 0/5) resulting in 0.22 studies with positive findings (min/max: 0/2) and 𝛼-investing found 0.48 (min/max: 0/8) positive findings and 0.37 (min/max 0/4) studies with positive findings; both of which are below the conservative baseline of 2. When the order is informed by the baseline findings, the sequential corrections procedures increase in the number of findings (findings [min/max]: 𝛼-debt: 3.49 [2/7], 𝛼-spending: 2.58 [1/4], 𝛼-investing: 3.54 [1/10]; and publications with positive findings [min/max]: 𝛼-debt: 2.38 [2/4], 𝛼-spending: 1.97 [1/3], 𝛼-investing: 2.54 [1/5]). All procedures now increase their number of findings above baseline. On average 𝛼-debt with a random order has a 19% increase in the number of published studies with positive findings, substantially less than the increase in the number of uncorrected studies. Two conclusions emerge. First, 𝛼-debt remains sensitive to the number of findings found regardless of the sequence of tests (fewer type II errors) and can never fall above the Bonferroni in regards to type II errors. At the same time, the other two sequential procedures can be more conservative than Bonferroni. Second, while 𝛼- debt does not ensure the false positive rate remains under a specific level (more type I errors), it dramatically closes the gap between the uncorrected and simultaneous number of findings.”

[In Methods section, emphasis added at relevant places]

"We then quantified the number of findings and the number of potential published studies with positive results that the different correction methods would present. *The number of findings is the sum of independent variables that were considered positive findings (i.e. p < 0.05, two-tailed). The number of potential studies that identify positive results is the number of dependent variables that had at least one positive finding. The rationale for the second metric is to consider how many potential non null-finding publications would exist in the literature if a separate group conducted each analysis."*

10) The authors used a real dataset to test a series of psychological hypotheses. They seem to have assumed that none of these hypotheses would pick up on any real effects in the data. Can they comment on the likelihood that their tests are establishing the true null distribution of effects, rather than actually being skewed by real effects? One solution would be to scramble the raw data in some way to ensure that even if there was a true effect, randomised and then processed voxels would not show this effect.

We do not believe we have made this assumption that there are no real effects in the data – we were quite agnostic about this. We do not make any judgment on what is a “real effect” but we contrast the consequences of different approaches and consider simultaneous correction to be a baseline (as that is an accepted consensus). We have however modified the statement below, when discussing the empirical examples, to make this even clearer:

"Importantly, in this evaluation of empirical data, we are not necessarily concerned with the number of true relationships with this analysis. Primarily, we consider the differences in the inferred statistical relationships when comparing the different sequential correction procedures to a baseline of the simultaneous correction procedures. These simultaneous procedures allow us to contrast the sequential approaches with current practices (Bonferroni, a conservative procedure, and FDR, a more liberal measure). Thus any procedure that is more stringent than the Bonferroni baseline will be too conservative (more type II errors). Any procedure that is less stringent than FDR will have an increased false discovery rate, implying more false positives (relative to the true positives).”

This shows (1) what convention (simultaneous correction) would produce as a result and we compare the consequences of no sequential correction and sequential correction.

The reviewer suggests scrambling the data, but this would be identical to the simulations, so we do not see any reason for adding them. Our logic of the argument is quite simple: (1) simulations, (2) an empirical example to show that the results are consistent with the simulations. The empirical example shows that our simulated effect actually has real world value. So we do not see a justification for adding the scrambled data analyses. However, we have added the following text when introducing the empirical example to clarify its purpose:

“We have demonstrated a possible problem with sequential tests on simulations. These results show that sequential correction strategies are more liberal than their simultaneous counterparts. Therefore we should expect more false positives if sequential correction methods were performed on a dataset. We now turn our attention to empirical data from a well-known shared dataset in neuroscience to examine the effect of multiple reuses of the dataset. This empirical example is to confirm the simulations and show that more positive findings (i.e. null hypothesis rejected) will be identified with sequential correction. We used 68 cortical thickness estimates from the 1200 subject release of the HCP dataset (Van Essen et al., 2012). All subjects belonging to this dataset gave informed consent (see Van Essen et al., 2013 for more details). IRB protocol #31848 approved by the Stanford IRB approves the analysis of shared data. We then used 182 behavioral measures ranging from task performance to survey responses (see Supplementary File 1). For simplicity, we ignore all previous publications using the HCP dataset (of which there are now several hundred) for our p-value correction calculation.”

11) The introduction and discussion could do a better job at contrasting different empirical traditions and statistical approaches. The introduction could make clearer that the current project assumes that most researchers are engaged in a particular (though dominant) type of research involving confirmatory hypothesis testing in which the goal is to explain rather than predict. Do you think the problem would be different if the focus was on prediction?The discussion mentions cross-validation and the problem with small sample sizes, but doesn't acknowledge explicitly the tension between explanatory and predictive research - Yarkoni & Westfall (2017) is a great reference on that topic: https://doi.org/10.1177/1745691617693393

We have added two new sections to help improve the focus of the article. These sections are: “Preliminary Assumption” (General response 1) in the Methods section and “An intuitive example of the problem” (General response 2) at the start of the article. This help contrast and hone in on the different statistical approaches. Further, we’ve made it clearer in the discussion section when we are talking about NHST and when we are talking about Bayesian statistics or prediction. For example, in the section entitled “Different perspective-specific solutions regarding sequential testing”, we have been more explicit that the reuseable held out data is a solution within the predictive framework and offered a longer introduction to predictive frameworks when introducing the problem. This paragraph now starts with:

"The second alternative is using held-out data within prediction frameworks. Instead of using statistical inference, this framework evaluates a model by how well it performs on predicting unseen test data (Yarkoni & Westfall, 2017). However, a well-known problem when creating models to predict on test datasets is overfitting. This phenomenon occurs, for example, if a researcher queries the test dataset multiple times. Reusing test data will occur when sequentially reusing open data. Held-out data on data repositories, as discussed above, is one potential solution here."

We have also added a solution about held out data on data repositories in the possible solution category.

12) FDR is not clearly defined and would need to be better justified given the strong limitations of such a concept, which Richard Morey and Deborah Mayo described as completely flawed:https://osf.io/ps38b/https://medium.com/@richarddmorey/redefining-statistical-significance-the-statistical-arguments-ae9007bc1f91

We have extended our the Methods section of the simultaneous correction strategies to include:

“We used the Bonferroni method and the Benjamini & Hochberg FDR method for simultaneous correction procedures (Benjamini & Hochberg, 1995). Both correction methods were run using multipy (v0.16, https://github.com/puolival/multipy). The FDR correction procedure intends to limit the proportion of type I errors by keeping in below a certain level. In contrast, Bonferroni error intends to limit the probability of at least one type-I error. Despite ideological criticisms and objections to both these methods (Bonferroni: (Perneger, 1998); FDR: (Mayo & Morey, 2017)), the Bonferroni correction is a conservative procedure that allows for more type II errors to occur and the FDR is a liberal method (i.e. allows for more type I errors). Together they offer a baseline range that allows us to contrast how the sequential correction procedures perform together.”

13) What if we are in a field in which inappropriate statistical methods are the norm: should future researcher using appropriate tools be penalised for analysing a dataset after many doomed attempts? You touch indirectly on the subject in the section "Gray-area when families are not clear". For instance, in a field dominated by fixed effect analyses of means, I would argue that researchers attempting to fit carefully justified generalised hierarchical models should be allowed to reset their alpha counter.

We believe we have addressed this point in General Response 1 (paragraph starting “types of analyses”) where we have highlighted what types of analyses we are considering with regards to data reuse that will require sequential correction. The example the reviewer gives here, which is a very good example about when sequential correction seems inappropriate which we have incorporated into the text.

14) The discussion mentions Bayesian statistics as a potential solution, but with the current trend in adopting arbitrary thresholds for Bayes factors, the same problems encountered in mindless frequentist practices will also apply to Bayesian/Laplacian ones:Gigerenzer, G. & Marewski, J.N. (2015) Surrogate Science: The Idol of a Universal Method for Scientific Inference. Journal of Management, 41, 421-440.

The reviewers are indeed correct that Bayes, applied poorly, is a problem. We have expanded the paragraph discussing Bayes statistics in a little more detail to explain why we have discussed it as an alternative framework, and included the reference suggested by the reviewers. The text now reads:

“The first alternative is Bayesian statistics. Multiple comparisons in Bayesian frameworks are often circumnavigated by partial pooling and regularizing priors (Gelman et al., 2013; Kruschke & Liddell, 2017). While Bayesian statistics can suffer from similar problems as NHST if misapplied (Gigerenzer & Marewski, 2014), it often deals with multiple tests without explicitly correcting for them, and may provide an avenue for sequential correction to be avoided.”